# Kernel Bioassay Evaluation of Maize Ear Rot and Genome-Wide Association Analysis for Identifying Genetic Loci Associated with Resistance to *Fusarium graminearum* Infection

**DOI:** 10.3390/jof9121157

**Published:** 2023-12-01

**Authors:** Jihai Zhang, Haoya Shi, Yong Yang, Cheng Zeng, Zheyi Jia, Tieli Ma, Mengyang Wu, Juan Du, Ning Huang, Guangtang Pan, Zhilong Li, Guangsheng Yuan

**Affiliations:** 1Yibin Academy of Agricultural Sciences, Yibin 644600, China; 2State Key Laboratory of Crop Gene Exploration and Utilization in Southwest China, Key Laboratory of Biology and Genetic Improvement of Maize in Southwest Region of Ministry of Agriculture, Maize Research Institute, Sichuan Agricultural University, Chengdu 611130, China

**Keywords:** maize, Gibberella ear rot, *Fusarium graminearum*, genome-wide association study, kernel bioassay

## Abstract

Gibberella ear rot (GER) caused by *Fusarium graminearum* (teleomorph *Gibberella zeae*) is one of the most destructive diseases in maize, which severely reduces yield and contaminates several potential mycotoxins in the grain. However, few efforts had been devoted to dissecting the genetic basis of maize GER resistance. In the present study, a genome-wide association study (GWAS) was conducted in a maize association panel consisting of 303 diverse inbred lines. The phenotypes of GER severity were evaluated using kernel bioassay across multiple time points in the laboratory. Then, three models, including the fixed and random model circulating probability unification model (FarmCPU), general linear model (GLM), and mixed linear model (MLM), were conducted simultaneously in GWAS to identify single-nucleotide polymorphisms (SNPs) significantly associated with GER resistance. A total of four individual significant association SNPs with the phenotypic variation explained (PVE) ranging from 3.51 to 6.42% were obtained. Interestingly, the peak SNP (PUT-163a-71443302-3341) with the greatest PVE value, was co-localized in all models. Subsequently, 12 putative genes were captured from the peak SNP, and several of these genes were directly or indirectly involved in disease resistance. Overall, these findings contribute to understanding the complex plant–pathogen interactions in maize GER resistance. The regions and genes identified herein provide a list of candidate targets for further investigation, in addition to the kernel bioassay that can be used for evaluating and selecting elite germplasm resources with GER resistance in maize.

## 1. Introduction

Maize is a cereal crop well adapted to many ecoregions, where a large proportion of people rely on it as their primary staple food. However, maize production and quality is often limited by fungal diseases, such as Gibberella ear rot (GER) caused by *Fusarium graminearum* (Schwabe) [1,2]. Serious yield losses can be caused by the disease, especially in the high temperate and humid regions in the world [3]. In southwest China, severe occurrences of GER in maize-growing areas cause yield loss [4]. Agronomic and chemical practices preventing the disease are currently insufficient when climatic conditions are favorable for the pathogen. The preferred method for controlling GER disease is to breed and cultivate resistant maize genotypes [5]. However, totally immune genotypes are not available, and commercial hybrids always have less resistance to the GER [6,7,8,9]. So, it becomes urgent to select resistant maize germplasm resources and identify broad-resistant genes for improving GER resistance.

*Fusarium graminearum* (*F. graminearum*) is a common fungal pathogen that infects many plant species, including maize, wheat, and rice [10]. In maize, the spores of *F. graminearum* could naturally infect maize upon conveyance through wind, rain splash, or insect infestation. Infected kernels are observed with a reddish-pink mold starting at the tip of a rotten ear [11]. Not only can the *F. graminearum*-caused disease reduce yield, but, also, the fungal infection produces diverse mycotoxins in the grain, including deoxynivalenol (DON) and zearalenone (ZEN), which threaten human and livestock health [3,5].

Accurate phenotypic assessment is the major bottleneck in identifying resistant genes and conducting genetics research on diseases [12]. A precise and convenient phenotypic evaluation method for GER resistance is challenging, owing to multiple factors influencing disease resistance scoring: inoculation time, inoculation method, and environmental conditions [13]. Under natural conditions, the unstable disease symptoms make it difficult to distinguish resistance differences among genotypes [14,15,16]. Therefore, development of a reliable phenotypic evaluation system is an essential step for improving breeding of maize resistance to *F. graminearum*. Most previous studies concerning phenotypic performance of maize ear rot were focused on the field evaluation [6,7,8,17,18,19,20]. Nevertheless, the field evaluation is time consuming, labor intensive, and influenced by numerous environmental factors. As an alternative method to precisely evaluate GER phenotypes, a kernel bioassay was developed for testing GER severity in laboratory [21,22,23]. Herein, healthy mature seeds were incubated with a fungal suspension to survey seed rot caused by occurring pathogens. The assay can be completely controlled under laboratory conditions to achieve more accurate phenotypic results, and it has been successfully applied in maize to evaluate the resistance to *Fusarium* spp. [24,25]. However, this method was based on a heavy workload and patience for calculating the number of spores, especially for a large population [21].

Previous efforts to characterize GER resistance indicated that the trait is generally considered as a quantitative inheritance with a complex genetic basis and is influenced by various environmental factors [5,12]. In recent years, much progress has been made in GER resistance, including the detection of quantitative trait loci (QTL), identification of resistant genes, and characterization of defense responses [6,7,8,9,13,14,15,16,17,18,19,26,27,28,29,30]. For instance, a previous study detected eleven QTLs, including a stable QTL, *qGER4.09*, conferring resistance to GER [23]. Although QTL mapping and omics research have identified a series of candidate regions or genes associated with disease resistance, the molecular mechanism of those genes underlying GER resistance have only been identified in a few cases due to inconsistent results from different populations and environments. To our knowledge, only limited research existed until recently on the genetic complexity of GER resistance and the identification of underlying genes, particularly for GER resistance in maize. Recently, the use of high-density single nucleotide polymorphism (SNP) datasets in combination with a genome-wide association study (GWAS) has emerged as a powerful alternative strategy to acquire target genes associated with desired traits [31]. This efficient approach can rapidly detect valuable natural variations in trait-associated loci as well as allelic variations in genes underlying quantitative and complex traits [32]. In maize ear rot, previous efforts were mainly focused on SNPs conferring resistance to *Fusarium verticillioides* ear rot (FER) [33,34,35,36], whereas there are very few reports on GER [5]. A previous study was conducted to detect significantly associated SNPs involved in resistance to maize GER [7]. However, no close-association SNPs were obtained in their research. In another study, a GWAS was performed in an association panel consisting of 316 diverse inbred lines and 10 co-localized association SNPs linked to GER resistance, including a peak SNP, and ten candidate genes were obtained [30].

In order to accurately estimate phenotypes of GER resistance, a maize association panel was evaluated for its resistance using the kernel bioassay. Then, we performed GWAS to identify SNPs and putative genes that may contribute to GER resistance. Moreover, we analyzed the alleles that can be potentially used to improve GER resistance for the next breeding programs. To our knowledge, this is the first time to dissect the genetic basis for maize GER resistance using a large-scale kernel bioassay indoors.

## 2. Materials and Methods

### 2.1. Maize Germplasm

The association panel containing core maize breeding materials from China, International Maize and Wheat Improvement Center (CIMMYT), and U.S.A. were evaluated for GER resistance using kernel bioassay in the laboratory. Detailed information regarding the plant materials has been reported in a previous study [30]. Due to seed availability, a total of 303 inbred lines were used for the experiment.

### 2.2. Experimental Procedure for Kernel Bioassay

For each line, mature seeds with similar sizes and shapes, without visible damage and without inoculation or infection, were selected for the experiment. To accurately evaluate the presence of the contaminating fungal infection, the healthy kernels of each line were surface disinfested with 70% ethanol for 2 min and then 0.6% sodium hypochlorite for 10 min and rinsed five times with sterile distilled water. To provide an infection court, the kernels were wounded by cutting the embryo side with a razor blade, with a cut depth of about 0.5 mm. Subsequently, the kernels were blotted dry and then placed in a 20 mL glass scintillation vial (diameter 2.8 cm and high 6.0 cm) with a screw cap (Wheaton Science, Head Biotechnology Co., Ltd., Beijing, China) and finally inoculated with 200 μL of a final concentration of approximately 1.0 × 10^5^ mL^−1^ spore suspension of *F. graminearum* (strain Fg 12002). The strain was isolated from naturally infected kernels using the single-spore isolation method and was kindly provided by Prof. P. Qi (Sichuan Agricultural University, China) for the present experiment. Control seeds were received an equal amount of sterilized distilled water. For each line, four seeds were placed into a vial as one biological replicate, with three replicates per experiment. The vials were kept in a humidity chamber under controlled conditions (27 ± 1 °C and 14:10 light/dark photoperiod). In view of no robust standards for kernel bioassay at present, we chose spore enumeration at different time points for the GER severity, according to the described study [21]. In current study, the vials were surveyed at 7, 14, 21, and 28 days after the *F. graminearum* inoculation for spore enumeration. Finally, the average number of spores across multiple time points was calculated to comprehensively evaluate the final GER severity of each line.

### 2.3. Phenotypic Analysis

Descriptive phenotypic analysis of the GER severity at different time points, including the range, mean, standard deviation (SD), and coefficient of variation (CV), were performed using the software SPSS 21.0 (http://www.spss.com, accessed on 29 September 2023). The SD was analyzed according to a one-way analysis of variance (ANOVA) model, and the CV (%) was calculated with 100 × SD divided by the mean. Due to strict controllable laboratory conditions, the effect of each line with environment interaction was absent, resulting in lack the analysis of the broad-sense heritability of GER.

### 2.4. GWAS Analysis

The association panels were genotyped, and detailed information on the population structure, kinship matrix, principal component analysis (PCA) matrix, and linkage disequilibrium (LD) decay distance have previously been described [37]. A total of 43,735 SNPs were obtained with a minor allele frequency (MAF) of less than 0.05 and a missing rate and heterozygosity greater than 20% for the present study. The average distance of LD decay was approximately 220 kb at *r*^2^ = 0.2. The LD attenuation distance around each SNP was used to search for candidate genes according to significant SNPs in GWAS. To identify SNPs with robust associations with GER, three models, the fixed and random model circulating probability unification model (FarmCPU), general linear model (GLM), and mixed linear model (MLM), were simultaneously applied in the GWAS analysis using the rMVP package in R software (version 1.0.7). Herein, the PCA and Kinship were added into the models for controlling false positive signals associated with traits [38]. The phenotype variance explained (PVE) of the SNPs was calculated as the following formula:$$r^{2}=\frac{\sum_{i=1}^{n}({ŷ}_{i}-{ŷ}_{}){}^{2}}{\sum_{i=1}^{n}(y_{i}-y_{}){}^{2}}$$

A Bonferroni test (0.05/N) was employed to select the SNPs significantly associated with GER, and a total of 24,535 (N) effective SNPs were ultimately obtained by the simpleM program in R [39]. Considering that the GER was determined by multiple minor effective genes, a moderated stringency threshold of −log (0.5/24,535) = 2.04 × 10^−5^ was chosen to determine the significant SNPs in FarmCPU and GLM. On the other side, a less stringent threshold of 1.0 × 10^−4^ for MLM was reasonable, according to previous reports [33,40]. Genes within the LD regions of significantly associated SNPs were considered as candidate genes governing GER resistance and then captured and annotated according to the B73 reference genome (B73 RefGen_v4) in the MaizeGDB database (https://www.maizegdb.org, accessed on 29 September 2023).

## 3. Results

### 3.1. Phenotypic Evaluation of GER Severity

The association panel consisting of 303 diverse inbred lines were evaluated for GER severity at 7-, 14-, 21-, and 28-days post-inoculation (dpi) using a kernel bioassay. The means of the spore values among the panel were 0.71 (7 dpi, ranging from 0.00 to 14.40 × 10^6^ mL^−1^), 2.77 (14 dpi, ranging from 0.00 to 64.00 × 10^6^ mL^−1^), 4.67 (21 dpi, ranging from 0.00 to 108.96 × 10^6^ mL^−1^), and 7.25 (28 dpi, ranging from 0.00 to 84.70 × 10^6^ mL^−1^), respectively (Appendix A). The outcomes of spore enumeration revealed that the number of spores increased gradually with the extension of the time-course inoculation (Table 1, Appendix A). Meanwhile, the vials observed that the hyphae also grew gradually during the inoculation process (Figure 1). Furthermore, their standard deviation (SD) and coefficient of variation (CV) were shown with high variation among the association panel (Table 1), implying that a high proportion of the phenotypic variation was exhibited during the *F. graminearum* infection. Overall, the spores’ number, hyphae growth, and variation were found with different types of response to *F. graminearum* infection, displaying that the laboratory-inoculated kernel method was available for quantifying fungal growth and biomass, evaluating the phenotypic performance in GER severity. The description on GER severity using a kernel bioassay in a large population was a first.

Considering the complexity of plant–pathogen interactions during *F. graminearum* infection, the final GER severity of each line was evaluated with an average number of spores across multiple time points. Based on the comprehensive evaluation, the top ten lines were obtained from the association panel with low spore concentration, including SCML1950 (0.48 × 10^5^ mL^−1^), Qi533 (0.52 × 10^5^ mL^−1^), JD7275 (0.74 × 10^5^ mL^−1^), End28 (0.81 × 10^5^ mL^−1^), 5Gong (0.83 × 10^5^ mL^−1^), CG698C102 (1.19 × 10^5^ mL^−1^), Su95-1 (1.28 × 10^5^ mL^−1^), Lin-1 (1.34 × 10^5^ mL^−1^), CLWN251 (1.41 × 10^5^ mL^−1^), and BJ005 (1.45 × 10^6^ mL^−1^) (Appendix A). The identified germplasms could be potentially utilized in a maize disease-resistance breeding program for GER. In general, these outcomes suggested that the kernel assay may provide a new way to screen resistant germplasm sources or detect specific loci associated with resistance to GER in maize.

### 3.2. Association Analysis and SNPs Discovery

According to the above comprehensive evaluation, the phenotypic data of 303 lines were ultimately obtained and further utilized to perform GWAS using the three models: FarmCPU, GLM, and MLM. In FarmCPU, only two significant association SNPs (PUT-163a-71443302-3341 and SYN9515) were identified with an adjusted threshold of 2.04 × 10^−5^, distributed on the chromosomes 1 and 2, respectively (Table 2, Figure 2a). In GLM, two SNPs (PUT-163a-71443302-3341 and PZE-110014176) were also found with significant association and localized on the chromosomes 1 and 10, respectively (Table 2, Figure 2b). For MLM, three significant association SNPs (PUT-163a-71443302-3341, SYN9515 and PZE-104154469) were detected with the threshold of 1.0 × 10^−4^, distributed on chromosomes 1, 2, and 4, respectively (Table 2, Figure 2c). The average PVE value of the identified SNPs was 5.6%, ranging from 3.51 to 6.42% (Table 2). Among them, the association SNP (PUT-163a-71443302-3341) had the greatest PVE value of 6.42%. A combination of the GWAS analysis from the three models revealed that only four significant SNPs were obtained associated with GER resistance. In detail, the SNPs PUT-163a-71443302-3341 on chromosome 1 and SYN9515 on chromosome 2 were repeatedly detected in different models, indicating that the two loci were reliable. Interestingly, the peak SNP (PUT-163a-71443302-3341) with the greatest PVE value was detected in all models (Figure 2), suggesting that the co-localized SNP was considered as a stable resistant locus. The peak region might contain important genetic components affecting GER resistance, and the useful SNP should be concerned for discovering candidate-resistant genes.

### 3.3. Genes Associated with GER Resistance

To identify genes with potential resistance to GER, candidate genes were explored within 220 kb upstream and downstream of the peak SNP (PUT-163a-71443302-3341). Finally, the focused SNP was adjacent to 12 putative genes according to the annotation information of the B73 reference genome (Table 3). Of these candidate genes, including the LRR-repeat protein and hydroxycinnamoyl transferase, several of them may have different roles in response to pathogen infection. Even though these genes were not described absolutely associated with GER resistance, the findings from this study suggest that the candidate SNP and linked genes should be taken into account and targeted to dissect functions involved in disease resistance.

### 3.4. Distribution of Favorable Alleles

As the ten elite resistant lines were obtained from the association panel via kernel bioassay evaluation, the favorable alleles of the association SNPs were estimated for further utilization. Herein, genotypes associated with a lower level of GER severity were defined as favorable alleles for GER resistance. The identified four association SNPs were distributed across the ten lines, and the favorable alleles percentages of these SNPs ranged from 10.0% to 90.0%, with two SNPs (PUT-163a-71443302-3341 and PZE-104154469) containing more than 50% favorable alleles, whereas the remaining two SNPs were no less than 50% (Figure 3). Moreover, each of the elite lines contained different favorable alleles, ranging from 1 to 4. In detail, the seven lines (5Gong, End28, SCML1950, JD7275, CLWN251, CG698C102, and Qi533) contained at least two favorable alleles, whereas the remaining three lines (Su95-1, Lin-1, and BJ005) contained only one favorable allele (Figure 3). Notably, the peak SNP (PUT-163a-71443302-3341) contained the nine most favorable alleles, suggesting that the significant SNP may have important effect on GER resistance.

## 4. Discussion

Disease severity caused by *F. graminearum*, particularly for GER, is often influenced by environmental conditions, host genotypes, and pathogenic races [5,12,26]. The establishment of a precise and convenient phenotypic evaluation method for GER resistance, capable of accommodating large populations, is a prerequisite for conducting genetics research in future [5,8]. Given the genotype and environmental interactions, field phenotypes of pathogen-caused diseases must be conducted in multiple environments over several years [12]. Thus, a reliable way for evaluating GER resistance is urgently necessary for eliminating external environmental factors. To achieve this goal, the GER severity of the maize association panel was evaluated through laboratory inoculation in the present study. This is the first large-scale phenotypic evaluation for GER severity indoors. Thus far, previous efforts on evaluating GER resistance for a large population were mainly on the field-inoculated test, and few efforts have been made indoors [7,26,30]. Although the indoor assay has been applied to assess disease severity for a few samples, not much progress has been reported on a large population due to the tedious process for spore enumeration at each inoculated time point [22,30,41].

In order to accurately describe GER severity, spore enumeration was performed across multiple time points during the *F. graminearum* infection. The results revealed a wide variation in GER severity among the lines (Table 1, Figure 1), suggesting that the indoor assay could effectively distinguish resistant variation for a large population. It should be noted that their SDs and CVs were showed with a big variation range among the association panel, demonstrating that the number of spores varied evidently at each time point. The most likely reason was that the assay found it difficult to control the seeds’ viability and nutrients, thus meaning that the *F. graminearum* could not produce spores steadily [13,21]. Furthermore, given no previous reports on resistant rating scales or grades for kernel bioassay, we assumed that the final GER severity should be evaluated comprehensively during *F. graminearum* infection. Indeed, the spores at each time point were only a partial reflection of the final resistance [13]. Thus, we considered the average spores across multiple time points as the phenotypic data for the final GER severity in the current study. However, this was just an initial tentative strategy in evaluating phenotypes of GER resistance for a large population. A further, prior approach on the phenotypic evaluation of GER severity through a kernel assay should be worthwhile. With the comprehensive evaluation, ten elite lines with low spore concentrations were obtained from the association panel, indicating that kernel assay offered an alternative way to evaluate phenotypes of GER resistance, and it could also accelerate to obtain resistant germplasms for improving resistance of maize GER disease. When compared with our previous study [30], only one inbred line, End28, was co-identified in both evaluations. The repeatability of the two methodologies was not very well. The possible reason is that different resistance mechanisms may be involved between the developing kernels and mature seeds during the infection process.

Plant resistance to pathogens is a complex interaction regulated by polygenic networks. In this study, GWAS was conducted by three models, FarmCPU, GLM, and MLM, to identify the candidate genomic regions and SNPs conferring GER resistance. Only four individual significant association SNPs with a range of PVE were identified in GWAS (Table 2, Figure 2). In a previous study, a total of 57 associated SNPs were obtained, and three interesting genes were identified conferring *Fusarium verticillioides* seed rot (FSR) resistance in their GWAS analysis [25]. That research provided much information on dissecting the genetic architecture for FSR resistance using the inoculated seed methodology. The results were different from previous similar studies [7,30]; fewer loci were obtained in the current study. The reason for this phenomenon largely relied on the different phenotypic data between the field and indoors. As mentioned earlier, most of the previous studies concerning phenotypic performances of GER severity were focused on the field evaluation, and the field phenotypic data varied widely, resulting in the detection of more variation loci conferring resistance [7,30]. On the contrary, the kernel assay was strictly controlled under laboratory conditions, thus leading to the phenotypic data varying more gently with a smaller variation than those in the field [21]. Despite the fact that fewer SNPs were obtained for our GWAS, two of them were repeatedly detected in different models, indicating that the identified loci were reliable to help understanding the complex genetic basis of GER resistance.

Furthermore, a combination analysis of GWAS results was performed to capture stable genomic region or key loci significantly associated with GER resistance. Interestingly, the peak SNP (PUT-163a-71443302-3341) with the greatest PVE value of 6.42% was co-localized in all models. Then, the significant SNP hit 12 specific genes, and several of them may have been involved in response to the pathogen infection (Table 3). For instance, a candidate gene Zm00001d032530 was annotated as LRR-repeat protein, which was widely reported to be involved in plant immunity [42]. Another candidate gene, annotated as hydroxycinnamoyl transferase (Zm00001d032527), was tightly associated with plant metabolism, playing an important role in the interaction between plants and pathogens [43]. In addition, ENSRNA049476973 and ENSRNA049476978, annotated as plant signal recognition particles (SRPs), were responsible for recognizing external pathogens in plant immunity [44]. The candidate gene Zm00001d032535 was annotated as a tetratricopeptide repeat (TPR)-like protein, playing an important role in regulation of growth and in response to environmental stimuli of plants [45]. Overall, further investigation on these candidate regions and the significant association of SNPs linked to candidate genes with potential resistance to GER in addition to the unknown genes is required. According to the distributions of the favorable alleles among the ten elite inbred lines, two SNPs were found containing more than 50% favorable alleles, suggesting that the SNPs should be emphasized in marker-assisted selections for a GER breeding program (Figure 3). Especially for the peak SNP (PUT-163a-71443302-3341), nine favorable alleles were observed across the ten lines. These findings implied that the detected alleles might have an important effect in response to GER resistance. In addition, the seven lines harboring more than two favorable alleles that exhibited low disease severity could be potentially utilized in maize disease-resistance breeding in the future.

## 5. Conclusions

In summary, we conducted a GWAS based on a laboratory-inoculated phenotypic evaluation to provide new and useful genetic information on maize GER resistance. We obtained four significant association SNPs through the GWAS, containing a peak significant SNP following 12 candidate genes. To our knowledge, this is the first large-scale GWAS focusing on the candidate regions and linked genes contributing to GER resistance by using a kernel bioassay indoors. These findings will help to better understand the genetic complexity of GER resistance mechanisms in maize.

## Figures and Tables

**Figure 1 jof-09-01157-f001:**
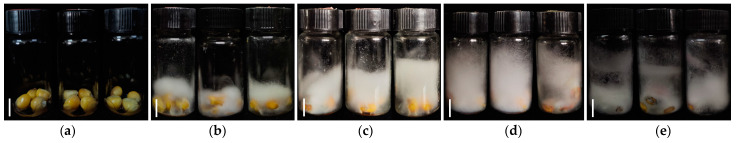
Typical phenotypic performance of kernels during the *F. graminearum* inoculation via the kernel bioassay in the laboratory. (**a**) Inoculated kernels at 0-day post-inoculation (dpi). (**b**) Inoculated kernels at 7 dpi. (**c**) Inoculated kernels at 14 dpi. (**d**) Inoculated kernels at 21 dpi. (**e**) Inoculated kernels at 28 dpi. Scale bar = 1 cm.

**Figure 2 jof-09-01157-f002:**
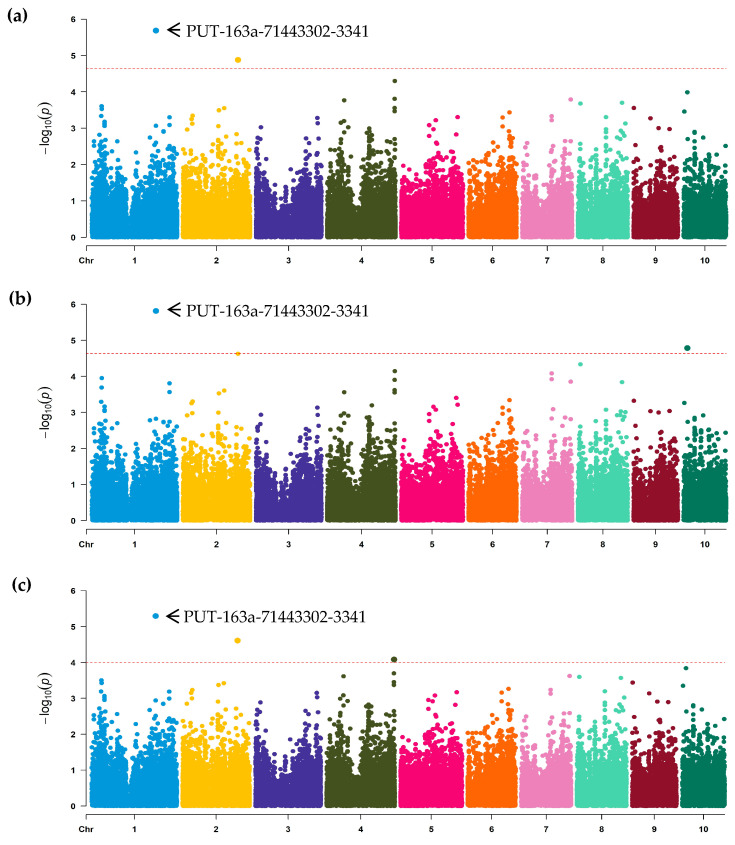
Manhattan plot of genome-wide association analysis (GWAS) for GER resistance with three models. (**a**) GWAS analysis using a fixed and random model circulating probability unification model (FarmCPU). (**b**) GWAS analysis using a general linear model (GLM). (**c**) GWAS analysis using a mixed linear model (MLM). The *Y*-axis value corresponds to -log_10_ (*p*) of *p*-value scores, and the *X*-axis indicates chromosomes and physical positions of SNPs. The red dashed lines show genome-wide significance at the adjusted thresholds of 2.04 × 10^−5^ for FarmCPU and GLM and 1.0 × 10^−4^ for MLM, respectively. The most significant association SNP PUT-163a-71443302-3341 was marked and co-localized by the combined FarmCPU, GLM, and MLM models.

**Figure 3 jof-09-01157-f003:**
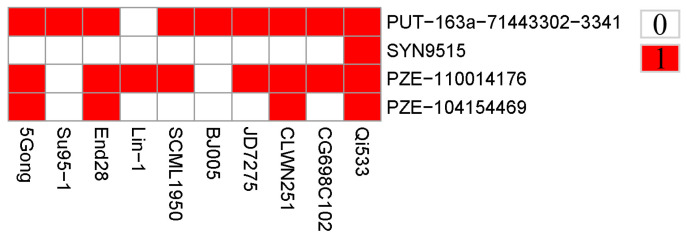
Distributions of favorable alleles among the ten elite lines. Red and white represent favorable and inferior alleles, respectively.

**Table 1 jof-09-01157-t001:** Phenotypic description among the association panel using the kernel bioassay.

Days Post-Inoculation (dpi)	Range (×10^6^ mL^−1^) ^a^	Mean ± SD (×10^6^ mL^−1^) ^b^	CV (%) ^c^
7	0.00–14.40	0.71 ± 1.47	207
14	0.00–64.00	2.77 ± 6.18	223
21	0.00–108.96	4.67 ± 9.38	200.8
28	0.00–84.70	7.25 ± 12.27	169

^a^ Range represents the number of spores with the concentration ×10^6^ mL^−1^. ^b^ Values are given as the mean ± SD (standard deviation). ^c^ The percentage of CV (coefficient of variation) represents the coefficient of phenotypic variation.

**Table 2 jof-09-01157-t002:** Significant association SNPs for GER resistance through GWAS with three models.

Model ^a^	SNP	Chr. ^b^	Position	Allele	MAF ^c^	*p* Value	PVE (%) ^d^
FarmCPU	PUT-163a-71443302-3341	1	226,136,399	G/A	0.29	2.06 × 10^−6^	6.42
FarmCPU	SYN9515	2	194,393,324	C/A	0.36	1.32 × 10^−5^	5.54
GLM	PUT-163a-71443302-3341	1	226,136,399	G/A	0.29	1.53 × 10^−6^	6.42
GLM	PZE-110014176	10	13,338,854	A/C	0.43	1.65 × 10^−5^	3.51
MLM	PUT-163a-71443302-3341	1	226,136,399	G/A	0.29	5.12 × 10^−6^	6.42
MLM	SYN9515	2	194,393,324	C/A	0.36	2.46 × 10^−5^	5.54
MLM	PZE-104154469	4	238,758,660	A/G	0.49	8.28 × 10^−5^	5.35

^a^ Model: FarmCPU, fixed, and random model circulating probability unification model. GLM, general linear model. MLM, mixed linear model. ^b^ Chr., chromosome. ^c^ MAF, minor allele frequency. ^d^ PVE, phenotypic variation explained.

**Table 3 jof-09-01157-t003:** The genomic regions of the peak SNP (PUT-163a-71443302-3341) and candidate genes associated with GER resistance ^a^.

Physical Position	Candidate Genes	Annotation
229515984-229517919	Zm00001d032527	hydroxycinnamoyltransferase13
229718687-229725690	Zm00001d032530	F-box/LRR-repeat protein
229758635-229765316	Zm00001d032531	Membrane steroid-binding protein 1
229766139-229766442	ENSRNA049476973	Plant signal recognition particle RNA
229816856-229817035	ENSRNA049476978	Plant signal recognition particle RNA
229829797-229830137	Zm00001d032533	--
229830451-229831175	Zm00001d022929	--
229831273-229832169	Zm00001d032534	--
229847655-229849510	Zm00001d022930	--
229847655-229849510	Zm00001d032535	Tetratricopeptide repeat (TPR)-like superfamily protein
229900429-229901097	Zm00001d032538	--
229992052-229992483	Zm00001d032542	plant/MXO21-9 protein

^a^ The relative physical positions on chromosome 1 were determined according to the B73 reference genome. Candidate genes were annotated in the region based on the B73 reference genome.

## Data Availability

Data are contained within the article.

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
