# Peer review of "Kernel Bioassay Evaluation of Maize Ear Rot and Genome-Wide Association Analysis for Identifying Genetic Loci Associated with Resistance to Fusarium graminearum Infection"

_jof, 2023, doi:10.3390/jof9121157_

Round 1
Reviewer 1 Report
Comments and Suggestions for Authors
see attachment.

Author Response
Dear Reviewer #1,
Thank you very much for your careful work and efforts with the many suggestions. We convinced that these valuable suggestions helped significantly to improve our text. We hope that the paper become better than the previous version was.
Comments to the Author
- The structure of the manuscript reads well, and no doubt the methodology for the GWAS was performed following the current standards and the methodology suggested for Maize Kernel Bioassay cited in reference 21.
Gibberella ear rot (GER) is an important fungal disease in maize cultivation, causing grain yield losses and contamination with mycotoxins. Identifying sources of resistance is not an easy task either in the field or in the Lab under-controlled environments. Therefore, it is a big effort to attempt to identify resistance genes among maize germplasms.
The present manuscript reports a different approach using the same population as described in reference No. 30 “Genome-wide association study discovers novel germplasm resources and genetic loci with resistance to Gibberella ear rot caused by Fusarium graminearum”.
When comparing the two approaches, only the accession End28 was identified under the two methodologies. Whereas the most resistant in the field test were not identified with the kernel bioassay. Therefore, I have doubts on the accuracy and repeatability of the kernel bioassay as authors mentioned in the manuscript “it is the is the first-time description on GER severity using kernel bioassay in large populations.” It would be desirable to compare and discuss both studies indicating the possibility that different resistance mechanisms are involved during the infection process at silking infection in the field and mature corn seeds.
Is the Kernel bioassay effective enough or reliable to identify sources of resistance? If not what the GWAS is doing is finding the presence of compounds in the grain which diminishes the seed contamination?
Response: Special thanks for your valuable comments. These comments are great significance to guide our research. We absolutely agreed with your opinion. Yes, identifying sources of resistance is not an easy task either in the field or in the lab under-controlled environments. In view of the complexity mechanism of GER, we tried to use different method to identify resistant germplasm resources for dissecting the genetic basis of maize GER resistance. To our knowledge, it is really the first time to describe GER severity by using kernel bioassay in large populations. In our previous report (reference No. 30), we identified several inbred lines with high resistance to GER based on field test. In this study, the same association panel were evaluated using kernel bioassay in lab. Unfortunately, only one inbred line was co-identified by two methods. The repeatability of the two methodologies was not very well. One hand, the kernel assay was performed under the controlled laboratory conditions, which was a new evaluated system for the GER severity, thus leading to the low repeatability among field-inoculated test and laboratory-inoculated evaluation. On the other side, as yor mentioned, the different developed stages of corn kernels may have different resistance mechanisms during the infection process, which may make the results of resistant phenotypes hard to remain consistent between the two methodologies. In our opinion, the kernel bioassay was a good tentative strategy in evaluating phenotypes of GER resistance for large population. So, the kernel bioassay and field test can be applied simultaneously to identify sources of resistance for GER and the both approaches are not conflict and even complement each other. The field test can be applied in vivo evaluation whereas the kernel bioassay could be used for in vitro experiment. We considered that the kernel bioassay is an effective and reliable strategy to identify sources of resistance and also could dissect the genetic analysis for GER. Then, the GWAS was performed to detect resistant loci for helping to understant the complex genetic basis of GER resistance.
- Authors should include seeds harvested from the field study in the seed bioassay without inoculation as is expected that some grains could be infected (not visible) to the naked eye. Please indicate if remanent seed were used in the bioassay experiment or seed harvested from the field, or what was the origin of the seed used in the experiment.
Response: Special thanks for your valuable comments. In our kernel bioassay experiment, the harvested seeds were used with uninfected healthy kernels. In addition, the experimental seeds were disinfested with ethanol and sodium hypochlorite and then rinsed with sterile distilled water for eliminating external factors. Following your suggestion, we have added more information to describe the experimental seeds in the section of materials and methods. (Please see the revised version).
- Please indicate if the fungus isolate in the present study was the same as the one used for field inoculation including pathogenicity test prior inoculation.
Response: Special thanks for your valuable comments. Yes, it is the same fungus isolate for the kernel bioassay indoor and field test. Following your suggestion, we have added the fungus isolate (strain Fg 12002) in the section of materials and methods. (Please see the revised version).
- 2.3. Phenotypic Analysis. Descriptive phenotypic analysis of the GER severity at different time points, including the range, mean, standard deviation (SD) and coefficient of variation (CV), were performed using the SPSS 21.0 software (http://www.spss.com) (accessed on 29 September 2023). Please include the phenotypic data per replication and means per entry. In a supplemental table if ANOVA was performed, please include the way the CV and SD were calculated.
Response: Thank you for your valuable comments. Following the suggestion, we have added a supplemental table to provide more detailed information for the phenotypic data of the GER severity at different time points. The calculated way of the CV and SD were added in the section of materials and methods. (Please see the Supplementary Materials Table S1).
- Due to strict controllable laboratory conditions, the effect of each line with environment interaction is absent, resulting to lack the analysis of the broad-sense heritability of GER. If that is the case, then why use replications? The use of replications with different randomization of the treatments allows to consider part of the environmental effects and to perform ANOVA.
Response: Thank you for your valuable comments. The good comments are great significance to guide our research. In our opinion, the replications were designed in present experiment to accurately evaluate the phenotypes of the GER severity, and we wanted to provided more detailed information about the kernel bioassay, including biological replications.
- Please compare and discuss your findings with other studies like Ju, M., Zhou, Z., Mu, C., Zhang, X., Gao, J., Liang, Y., et al. (2017). Dissecting the genetic architecture of Fusarium verticillioides seed rot resistance in maize by combining QTL mapping and genome-wide association analysis. (Sci. Rep.7:46446. doi: 10.1038/srep46446).
Response: Thank you for your valuable comments. Special thanks for your kind help and providing the important reference to us. Following the suggestion, we have cited the report and added more information in the discussion section. (Please see the revised version).
- Among the SNP markers found in the kernel bioassay, the SNP (PUT-163a-23 71443302-3341) was identified with the greatest PVE value, was co-localized in all models. Whereas in the field study was the SNP PZE-105079915. Please indicate and punctuate strong evidence that the current study is as reliable as the field test and show advantages and disadvantages.
Response: Thank you for your valuable comments. In many previous reports, the maize GER resistance is a complex quantitative trait caused by multigenes and influenced by environmental conditions, host genotypes and pathogenic races. As mentioned earlier, the different developed stages of corn kernels may have different resistance mechanisms during the infection process, which may make it hard to obtain consistent SNP for the quantitative resistance between the field-inoculated test and laboratory-inoculated evaluation. The phenotypes from the kernel bioassay were compatible with field phenotypic test, which could be considered as in vivo evaluation and in vitro experiment, respectively. In our opinion, the two SNPs (PUT-163a-23 71443302-3341 and PZE-105079915) were considered reliable loci for the GER resistance by the different phenotypic data. Therefore, the current study is a good attempt to identify resistant genes or SNPs among maize germplasms. The current study has several advantages. First, the GER severity of large population can be performed at any time indoor, do not need to consider environmental temperature. Second, the phenotypes of GER were more accurate due to no external environmental factors. Third, the kernel bioassay in the current study is a good strategy and could complement to field test for comprehensively evaluating GER severity. Of course, there are some disadvantages among the current study. For instance, the phenotypes of GER severity were derived only from lab, and may not correspond to the natural environment. On the other side, few loci were obtained due to strict controllable laboratory conditions. In addition, the indoor evaluation is very tedious process and labor-intensive.
- Due to strict controllable laboratory conditions, the effect of each line with environment interaction is absent. Growing fungi under the laboratory just by the position on the bench and the amount and type of light received will cause variation in conidiation. Therefore, the petri plate assay is in my opinion more reliable than the glass scintillation vial.
Is the experimental unit and the seed sample size large enough to determine the genetic variability for each line (12 seeds). In the petri plate experiments the experimental unit is constituted by 10 kernels.
Response: Thank you for your valuable comments. The comments are great significance to guide our research. We absolutely agreed with your opinion. Yes, growing fungi could be influenced by the laboratory conditions. Following the suggestion, we will try to use the petri plate assay in our laboratory and expect more better results.
Reviewer 2 Report
Comments and Suggestions for Authors
The authors conducted a genome-wide association in Giberrela ear rot using kernel laboratory assay as a study model. Spore concentration was correlated with the kernel severity, and a few SNPs were considered as stable resistant loci. Some of these genes should linked to GER resistance.
L71-L90: Evidence in the literature indicates that the infection mechanisms of Fusarium in the ear are similar between different species. On the other hand, this biological model has various reports of SNPs. I suggest that the authors include an overview of the associated genes. and its potential implications in the resistance mechanism.
L166: word correction “infection”.
L318-The most critical part of the research is the biological implications of the genes associated with the GER resistance mechanism. This section is poorly addressed.
Author Response
Dear Reviewer #2,
Thank you very much for your careful work and efforts with the many suggestions. We convinced that these valuable suggestions helped significantly to improve our text. We hope that the paper become better than the previous version was.
Comments and Suggestions for Authors
- The authors conducted a genome-wide association in Giberrela ear rot using kernel laboratory assay as a study model. Spore concentration was correlated with the kernel severity, and a few SNPs were considered as stable resistant loci. Some of these genes should linked to GER resistance.
L71-L90: Evidence in the literature indicates that the infection mechanisms of Fusarium in the ear are similar between different species. On the other hand, this biological model has various reports of SNPs. I suggest that the authors include an overview of the associated genes. and its potential implications in the resistance mechanism.
Response: Special thanks for your valuable comments. The comment is great significance to guide our research. Following the suggestion, we had added more previous reports conferring GER resistance in the introduction section. (Please see the revised version).
- L166: word correction “infection”.
Response: Special thanks for your careful review. Following the comment, corrected as suggested. (Please see the revised version).
- L318-The most critical part of the research is the biological implications of the genes associated with the GER resistance mechanism. This section is poorly addressed.
Response: Special thanks for your valuable comments. We absolutely agreed with your opinion. Following the suggestion, we had added more previous reports to demonstrate the identified genes associated with plant resistance in the sections of introduction and discussion. (Please see the revised version)
Reviewer 3 Report
Comments and Suggestions for Authors
The manuscript “Kernel Bioassay Evaluation of Maize Ear Rot and Genomewide Association Analysis for Identifying Genetic Loci Associated with Resistance to Fusarium graminearum Infection” is a very interesting study. Just a few recommendations to the authors:
1. What type of Fusarium spores are the authors referring to throughout the manuscript? It is recommended to define it in one of the sections.
2. In the Abstract section, line 14, the phrase “and contaminates several potential mycotoxins” is not very clear, the authors are recommended to rewrite it.
3. In the Introduction section, lines 86-87, also the sentence “Until recently……..to maize GER” is not very clear, the authors are recommended to rewrite it.
4. In Materials and Methods section, it is recommended that the authors specify the strain and origin of Fusaium graminearum used; also, the pathogenic race because they comment on the pathogenic races in the discussion section. Likewise, it is recommended to specify the size of the jars and type of lid used.
5. In the Results section, lines 184-191, the authors describe low values in the concentration of spores in ten lines; but in Table 2 the spore concentration ranges start at zero, could the authors explain this? In Figure 1, there is a missing space between the word "Figure" and the number "1", and a measurement scale is missing. Authors are also recommended to include a photo of the controls on each date, and also photos of the spores on each date, if available.
Author Response
Dear Reviewer #3,
Thank you very much for your careful work and efforts with the many suggestions. We convinced that these valuable suggestions helped significantly to improve our text. We hope that the paper become better than the previous version was.
Comments and Suggestions for Authors
The manuscript “Kernel Bioassay Evaluation of Maize Ear Rot and Genomewide Association Analysis for Identifying Genetic Loci Associated with Resistance to Fusarium graminearum Infection” is a very interesting study. Just a few recommendations to the authors:
- What type of Fusarium spores are the authors referring to throughout the manuscript? It is recommended to define it in one of the sections.
Response: Special thanks for your valuable comments. We are sorry for the missing information. Following your suggestion, we have added the fungus isolate (strain Fg 12002) in the section of materials and methods. (Please see the revised version).
- In the Abstract section, line 14, the phrase “and contaminates several potential mycotoxins” is not very clear, the authors are recommended to rewrite it.
Response: Special thanks for your careful review. Following the comment, we have modified the sentence as suggested. (Please see the revised version).
- In the Introduction section, lines 86-87, also the sentence “Until recently……..to maize GER” is not very clear, the authors are recommended to rewrite it.
Response: Special thanks for your careful review. Following the comment, we have modified the sentence as suggested. (Please see the revised version).
- In Materials and Methods section, it is recommended that the authors specify the strain and origin of Fusaium graminearum used; also, the pathogenic race because they comment on the pathogenic races in the discussion section. Likewise, it is recommended to specify the size of the jars and type of lid used.
Response: Special thanks for your valuable comments. Following your suggestion, we have added the fungus isolate (strain Fg 12002) and detailed size of the glass scintillation vial in the section of materials and methods. (Please see the revised version).
- In the Results section, lines 184-191, the authors describe low values in the concentration of spores in ten lines; but in Table 2 the spore concentration ranges start at zero, could the authors explain this? In Figure 1, there is a missing space between the word "Figure" and the number "1", and a measurement scale is missing. Authors are also recommended to include a photo of the controls on each date, and also photos of the spores on each date, if available.
Response: Special thanks for your valuable comments. In Table 2, the spore concentration ranges start at zero. The possible reason is that the pathogen was not produce spores in the vegetative growth stage, leading to zero number at that time point. Another possible reason is that the grain nutrients of some inbred lines were exhausted rapidly and no longer produced spores at that point. This may happen in any time point during inoculation in the kernel bioassay. (Please see the Supplementary Materials Table S1). In additon, the Figure 1 was corrected as suggested and a scale bar was added in the Figure. Moreover, because the kernels in vials had basically germinated and rotted after 7 dpi, the control vials were disposed of in time during the experimental process, so we did not keep the photos of the controls on each date. We are sorry for the negligence. On the concerns for the spores in the kernel bioassay, we would like to provide a supplemental Figure to demonstrate the details of the spores on each time point during inoculation. (Please see the Supplementary Materials Figure S1).
Round 2
Reviewer 1 Report
Comments and Suggestions for Authors
Author Response
Dear Reviewer #1,
Thank you very much for your careful work and efforts with the many suggestions. We convinced that these valuable suggestions helped significantly to improve our text. We hope that the paper become better than the previous version was.
Comments to the Author
- My major concern is to discuss and show evidence that the spore enumeration is accurate or not to perform the GWAS study.
In my opinion spore numbers are not strong enough due to the variability existing in the bioassay and inconsistency on the respective dates of measurement. GER severity at 7-, 14-, 21- and 28-days postinoculation (dpi) using kernel bioassay.
Response: Special thanks for your valuable comments. These comments are great significance to guide our research. In the kernel bioassay, the spores at different time point were calculated carefully for each lines. As mentioned earlier, this approach is a good tentative strategy in evaluating phenotypes of GER resistance for large population. In view of no robust standards for kernel bioassay at present, we chosen spore enumeration at different time pionts for the GER severity according to the described study [21].
We absolutely agreed with your opinion that the spore enumeration could be variability among the respective dates of measurement. To reduce this variability and increase phenotypic accuracy, four time points were set for spore enumeration in the bioassay. By this way, the GER severity of the association panel could be evaluated as accurate as possible by using kernel bioassay. Although there are some imperfections in current bioassay, a different method to identify resistant germplasm resources and dissect the genetic basis of maize GER resistance is worthwhile.
- In the following statement you indicated the concentration of spores per ml: SCML1950 (0.48 × 106 ml-1), Qi533 (0.52 × 106 ml-1), JD7275 (0.74× 106 ml-1), End28 (0.81× 106 ml-1), 5Gong (0.83× 106 ml-1), CG698C102 (1.19 × 106 ml-1), Su95-1 (1.28 × 106 ml-1), Lin-1 (1.34 × 106 ml-1), 187 CLWN251 (1.41 × 106 ml-1) and BJ005 (1.45 × 106 ml-1). Whereas in the supplemental table you indicted
Material name |
7 DPI (×105) |
14 DPI (×105) |
21 DPI (×105) |
28 DPI (×105) |
Mean (×105) |
Response: Special thanks for your careful review. We sincerely apologize for the mistake. In the current kernel bioassay, the final concentration of F. graminearum is approximate 1.0 × 105 ml−1 spores suspension for indoor inoculation. The high concentration of F. graminearum with 1.0 × 106 ml−1 spores suspension was used for field test. We didn't notice the change at first. For field test, higher concentration of the spores is required to obtain wide variation of resistance on GER severity. And for indoor evaluation, a moderate concentration of the spores was deemed suitable to evaluate the resistant phenotypes for GER. To survey the produced spores at different time points, the kernels were needed to maintain a certain amount of nutrients avoiding rapid exhaustion during the long time inoculation. So, slightly lower concentration was selected for the current kernel bioassay. Following the comment, the concentrations of the initial inoculation and identified resistant germplasms were corrected. (Please see the new revised version).
Once again, special thanks for editor’ and reviewer’ valuable comments. We hope that those changes in the revised manuscript have satisfactorily answered your concerns.
Best wishes,
Zhilong Li
Yibin Academy of Agricultural Sciences, Yibin, China
Guangsheng Yuan
Sichuan Agricultural University, Chengdu, China
Round 3
Reviewer 1 Report
Comments and Suggestions for Authors
At this point I do not have a further comment to the manuscript. Please contact a Giberella zea specialist to help in your final decision.